# PAX Family, Master Regulator in Cancer

**DOI:** 10.3390/diagnostics15111420

**Published:** 2025-06-03

**Authors:** Erica Giacobbi, Maria Paola Scioli, Francesca Servadei, Valeria Palumbo, Rita Bonfiglio, Pierluigi Bove, Alessandro Mauriello, Manuel Scimeca

**Affiliations:** 1Department of Experimental Medicine, TOR, University of Rome “Tor Vergata”, 00133 Rome, Italy; erica.giacobbi@gmail.com (E.G.); sciolimp@hotmail.it (M.P.S.); francescaservadei@gmail.com (F.S.); valeria.palumbo@uniroma2.it (V.P.); rita.bonfiglio@uniroma2.it (R.B.); 2Department of Surgical Sciences, University of Rome “Tor Vergata”, Via Montpellier 1, 00133 Rome, Italy; pierluigi.bove@uniroma2.it

**Keywords:** PAX family, cancer, EMT, cell death, therapy

## Abstract

PAX genes, known as master regulators, encode paired box (PAX) proteins that govern key processes in organ development and are widely expressed in normal tissues. Notably, PAX proteins also play a pivotal role in both promoting and suppressing tumorigenesis. They influence essential cellular functions such as survival, proliferation, fate determination, differentiation, invasion, metastasis, and the formation of oncogenic fusion proteins. In this review, we summarize the current understanding of these transcription factors. First, we provide a brief overview of their molecular structure, which underlies their classification into four subgroups. Then, we examine the expression patterns of each PAX gene across organ systems and explore their biological roles in the most relevant malignant neoplasms affecting human health. Additionally, we highlight their diagnostic, prognostic, and predictive significance in the context of cancer.

## 1. PAX Structure, Subgroups and General Functions

PAX genes encode a family of tissue-specific transcription factors, found in both vertebrates and invertebrates. They were called master regulator paired box (PAX) proteins, due to their ability to regulate gene transcription during different mechanisms that control embryogenesis, tissue, and organ development.

PAX proteins are characterized by the presence of three conserved elements: two DNA-binding domains, the paired domain (PD) and homeodomain (HD), and the short octapeptide sequence (OP) [1,2].

The first studies conducted on this family started in the late 1980s and the paired domain was first identified in *Drosophila melanogaster*. The importance of these paired domain transcription factors quickly became apparent, as mutations in these genes were associated with developmental changes.

The paired domain is composed by 128 amino-acids and binds specific DNA sequences. It presents three alpha helices linked, thanks to the octapeptide sequence, to the homeodomain, composed by 61 amino-acids and presenting three alpha helices [3].

Given their structural similarities, nine paired box genes are known in mice (Pax1 to Pax9) and in human beings (PAX1 to PAX9), divided into four subgroups based on their structural similarities, such as on the presence or absence of the octapeptide (OP) and a complete or truncated version of the homeodomain (HD). The structural similarity mirrors some activities and functions.

PAX genes from subgroups II (PAX2, PAX5, and PAX8) and III (PAX3 and PAX7) are involved in essential processes such as cell survival, motility, and tumor progression. The involvement of members from subgroup I (PAX1 and PAX9) and IV (PAX4 and PAX6) in cancer processes seem less significant.

I (PAX1 and PAX9)II (PAX2, PAX5, PAX8)III (PAX3 and PAX7)IV (PAX4 and PAX6)

PAX proteins from groups II (PAX2, PAX5, PAX8), III (PAX3 and PAX7), and IV (PAX4 and PAX6) contain a truncated or full homeodomain (HD), which allows them to bind DNA and regulate specific subsets of target genes.

PAX proteins from groups I, II, and III also contain an octapeptide linker, whose function is to bind specific cofactors required to control the transactivation of the PAX proteins in groups II and III.

PAX proteins from group I (PAX1 and PAX9) lack the HD, which suggests that the HD domain is not essential for the DNA binding function. In fact, this domain promotes the interaction between the transcription factor and DNA through the PD [4,5].

The PD is highly conserved in the protein structure, indicating that PAX proteins interact with DNA through similar binding motifs. However, each individual PAX protein regulates specific target genes, and this occurs through alternative splicing, which generates different mRNAs, each coding for a different protein.

By deepening the study of PAX molecules, it becomes evident that PAX proteins are essential for normal cellular development, but their dysregulation can result in significant consequences for cellular behavior and contribute to both cancer initiation and progression (see Table 1).

It is well known that embryological development and the transformation of a cell into a malignant neoplastic cell are parallel processes. Aberrant activation of oncogenic pathways is the result of damage to existing molecular pathways, fundamental for normal tissue development.

It has been observed that PAX molecules, widely expressed in many normal tissues, play a central role in many of the key mechanisms that promote or suppress tumorigenesis.

The molecular mechanisms through which PAX proteins influence cancer behavior involve several key processes, such as cell survival and proliferation, cell fate and cell differentiation, invasion and metastasis. It underscores their importance in cancer biology and offers valuable insights into the understanding and in finding potential treatment of human diseases.

Strating from these considerations, this review aims to explore the mechanisms by which the expression of PAX molecules contributes to tumorigenesis, as well as their impact on diagnosis and prognosis in some of the most significant neoplasms. To this end, we revised the role of PAX proteins in selected malignant neoplasms, with a focus on the cancers where their oncogenic functions have been most extensively characterized.

## 2. PAX5 in B-Cell Maturation and in Hematological Malignancies

PAX5 (paired box 5), also known as BSAP (B-cell-specific activator protein), is the only PAX molecule that plays a role in the management of hematopoietic diseases. Its specific role in B-cell maturation makes PAX5 a highly specific marker of the B-cell lineage and incorporating it into a panel of immunohistologic markers for investigating undifferentiated neoplasms provides diagnostic benefits.

However, PAX5 expression has been found in some T-lymphomas, myeloid neoplasms and in a small percentage of undifferentiated non-hematolymphoid tumors. This evidence has prompted further investigation into the role of PAX5 in tumorigenesis [38].

The B cell commitment and differentiation of common lymphoid progenitors (CLPs) to terminal plasma cell differentiation depend on PAX5 transcriptional function that mediates development towards the mature B-cell stage by silencing genes inappropriate for B-cell development and enhancing lineage-specific B-cell genes.

On one hand, PAX5 promotes up-regulation of genes involved in BCR complex formation and signaling, such as Igα and CD79, essential components of the complex, and CD19 and CD 21, costimulatory receptors of BCR [39].

On the other hand, PAX5 inhibits the expression of genes involved in myeloid plasticity or plasma cells differentiation, such as Fms-Like Tyrosine kinase-3 (Flt-3), NOTCH receptor-1 (mediator of T-cell differentiation, or B-cell Induced Maturation Protein-1 (BLIMP-1) [40].

According to its role as tailor of B cell commitment process, PAX5 acts as a regulator of the VDJ recombination [41]. Indeed, in PAX5-deficient pro-B cells, recombination of the distal VH gene fails, and the ectopic expression of PAX5 overcomes the defect [42]. Moreover, together with EBF1, PAX5 indirectly regulates Igh locus by repressing the cohesin-release factor gene Wapl, involved in loop extrusion of VH genes and thus in VH-DJH recombination [43].

Genetic modulations involving PAX5 gene are also reflected in the molecular, clinical and pathological heterogeneity of B-cell-derived lymphomas and leukemia at various degrees of differentiation [21].

### 2.1. PAX5 in Precursor B-Cell Neoplasm

B-lymphoblastic leukemia/lymphoma (B-ALL/LBL) is the most common pediatric malignancy that develops from the precursor lymphoid cells committed to the B-cell lineage. Blocked at an early stage of differentiation, tumor lymphoid cells are able to involve bone marrow and blood, arriving at the extramedullary sites [44].

In B-ALL, PAX5 genetic alterations are a hallmark and characterize 31% of cases, involving deletions, translocation, and point mutations, and result in dedifferentiation of pro-B cells and aggressive lymphomagenesis [45,46].

#### 2.1.1. PAX5 Deletions

Concerning deletions of PAX5, monoallelic mutations have been observed in about 30% of children with B-ALL [47]. The deleted region could be wide, involving other genes or entire part of the chromosome 9, or be directly associated with the gene sequence [48]. These genetic alterations determine the loss of PAX5 protein expression or the production of a protein lacking the DNA binding domain and/or transcriptional regulatory domain.

Frequently, deletions of PAX5 represent secondary mutations, acquired after primary oncogenic mutations of genes involved in genome instability [47,49]. This leads to a complex genomic profile and rearrangements (such as translocation related to BCR-ABL1 [50]), which commonly accompany PAX5 deletions.

In mice, biallelic deletions of PAX5 have been shown to completely block B cell differentiation and induce premature death [51].

#### 2.1.2. PAX5 Translocation

With a quite small rate of occurrence, translocations of Chr9 involving PAX5 genes represent 2.5% of pediatric and 1% of adult B-ALL patients [49]. There are many partners which participate in the protein fusion formation with PAX5 [52].

Specifically, most of these fused proteins retain the DNA-binding domain, while losing the activating C-terminal domain and act as dominant-negative proteins to interfere with the wild-type (WT) PAX5 [53]. One of the most frequent gene fusion found in B-ALL is PAX5/ETV6, due to t(9;12)(p13;p13) [54]. ETV6 is one of main transcription factors associated with B-cell development and acts directly on PAX5 through the recruitment of SIN3A and HDAC3 on its locus [55].

The PAX5/ETV6 fusion protein acts as a promoter of leukemic B-cell activation, unlike deletions or point mutations that result in an inactivated form of PAX5, classifying it as a tumor suppressor.

In vitro evidence of B-cell progenitors indicates a role of PAX5/ETV6 in reducing CD19, *BLNK*, *MB-1*, *FLT3*, and μ heavy chain expression, thus blocking B-cell differentiation, enhancing migratory ability through CXCR12, and increasing survival [56].

Moreover, PAX5/ETV6 cells have an augmented activation of Lck, followed by the over-activation of STAT5 signaling pathway (which involves c-Myc and Ccnd2) [57]. The over-activation of Lck makes this type of genetic alteration of PAX5 (included in Philadelphia chromosome-like (Ph-Like) B-ALL) possibly eligible for specific target therapy, such as Lck-inhibitors [58].

Because of that, PAX5/ETV6 is considered an oncogenic protein, promoting leukemic transformation [59].

Frequently involved in translocation with chromosome 9 is the pericentromeric region of 7q, where *ELN*, *AUTS2* [60], and *POM121* [61] are located and form a cluster of PAX5 partner genes [62]. PAX5/ELN (elastin) has been identified in some cases of pediatric B-ALL.

The related translocation of t(7;9)(q11;p13) causes the generation of a fusion protein that retains the ability to bind DNA, specifically the promoter of those genes whose expression is regulated by PAX5 [63].

Here, the fusion protein PAX5/ELN may act as dominant-negative compared to WT PAX5. However, other possible mechanisms could be associated with leukemic transformation of B-cells progenitors, when PAX5/ELN occurs. For instance, an in vivo model showed that heterozygous mice carrying the PAX5/ELN gene partially alter the expression of genes regulated by PAX5, but it was associated with recurrent secondary gene mutations such as Kras, Pax5, and Jak3 [64]. The result is a modification of relative pathways required for cell proliferation.

Depending on fusion partner genes, PAX5-fused proteins could have different functions. PAX5/JAK2 is a nuclear DNA-binding JAK2 fusion protein with kinase activity [65] belonging to the Ph-like ALL subgroup [66,67]. Jak2 and Pax5 genes share the same chromosome, so a PAX5/JAK2 fusion gene could be determined by several rearrangements such as t(9;9)(p13;p24), del(9)(p13p24) or inv(9)(p13p24) [68].

This type of alteration, generally heterozygous, does not interfere with PAX5-regulated genes but acts on increasing STAT5 phosphorylation directly within the nucleus and its pathway after leukemic transformation [69,70]. Furthermore, t(9;15)(p13;q24) results in an in-frame fusion of PAX5 to the promyelocytic leukemia (PML) gene. This altered form of PAX5 has been found in some cases of childhood B-ALL and implies the generation of a protein that retains the paired domain, the octapeptide and the partial homeodomain of PAX5, and the whole PML protein [71].

Therefore, studies support the impact of PAX-JAK2 on leukemia development and suggest a potential therapeutic role for patients with leukemia subtypes carrying this molecular signature [70].

#### 2.1.3. PAX5 Point Mutation

Point mutations within PAX5 generally affect the DNA-binding domain, impairing its ability to bind DNA and transcriptionally activate target genes [48]. One of the most frequent and well-characterized point mutations within DNA binding domain is PAX5 P80R [49].

This somatic mutation has been observed with a frequency of 0.4% in standard ALL and 3.1% in high-risk pediatric B-cell precursor (BCP) [72,73].

The P80R mutation implies the substitution of a Proline with an Arginine within DNA binding and determines the ability to bind different sequences, modulating different genes [74]. P80R binds to an intragenic promoter of MEGF10, a marker gene uniquely activated by PAX5 P80R [46].

PAX5 P80R blasts are arrested at the pre-pro-B stage and often can undergo a phenotype switch to a monocytic lineage [75].

Because of different genetic alterations and distinct gene expression profile (GEP), PAX5 P80R is considered a B-ALL subtype [67,72,76], with a better outcome compared the other ones [77].

PAX5 deletions or other inactivating rearrangements act as co-activator for malignant cell transformation. Several mutated genes have been identified as primary activators for B-cell progenitor malignant transformation, and together with PAX5 increase the vulnerability and the frequence of B-ALL occurrence [78]. For instance, in mice, the double heterozygosis for Ebf1 and Pax5 induce an increased c-Myc and STAT5 expression through IL-7 pathway activation [79]. Ebf-1 represents the up-stream activator of PAX5 and c-Myc, which is in turn down-regulated by PAX5 [80].

This loop allows a proper regulation of c-Myc activity and stimulation of pre-B cells, avoiding their leukemic transformation. Increased activation of IL-7/Il-7r drives STAT5 activation, which in turn promotes PAX5 expression in normal B-cell development [81,82]. However, when both Ebf1 and Pax5 results were deleted or inactive, increased up-regulation of IL-7 increase malignant transformation by up-regulating the glycolytic and folate metabolism and promoting B-cell progenitor plasticity [79].

#### 2.1.4. Germline Mutation

Although rare, germline mutations can be found in some cases of B-ALL. These mutations are generally not unique and are not directly inducing B-ALL, even if they increase the susceptibility of carriers to this disease. Indeed, other somatic mutations are required as the leukemic transformation happens. Among the germline mutations, *PAX5* missense variant c.547G>A (p.Gly183Ser) determines a mutation in exon 6 corresponding to the region coding for the octapeptide domain [83]. This mutation has been identified in four unrelated children and was found to segregate with disease in two unrelated children [83] and is associated with a reduced PAX5 activity as transcription factor [84]. Other germline mutations of PAX5 include heterozygous germline variants, such as c.113G>A (p.Arg38His), which affects the paired domain of the protein [85], a deletion of exon 6 [86], or *PAX5* c.547G>C (p.Gly183Arg) [87].

All these germline variants confer susceptibility for childhood leukemia, due to an impairment of B-cell differentiation in the bone marrow and a reduction in mature memory B cells in the peripheral blood [88].

A different mechanism associated with germline mutation is related with the heterozygous variant in *PAX5*, a single nucleotide deletion resulting in a reading frameshift PAX5 c.548delG [89]. In this case, despite the frameshift variant being predicted to cause a premature arrest of the protein synthesis, it is likely that its mRNA may be degraded through the nonsense-mediated mRNA decay pathway [89]. This mechanism would therefore cause a *PAX5* germline haploinsufficiency instead of a protein variant, as in other families.

We can therefore conclude that acute lymphoblastic leukemia (ALL), particularly its B-progenitor lineage (B-ALL), is closely linked to genetic alterations in PAX5. Pax5 mutations, including deletions, translocations, and point mutations, play significant roles in disrupting normal lymphoid differentiation, leading to malignant transformation. Understanding these mechanisms offers critical insight into disease pathogenesis and potential therapeutic interventions. Targeted therapies addressing specific PAX5 alterations, such as STAT5 or Lck inhibitors, hold promise for improving outcomes in B-ALL patients with distinct molecular profiles.

### 2.2. PAX5 in Mature Cell Lymphoid Neoplasm

Evidence of PAX5 involvement in lymphomas paints it mainly as secondary player in pathogenic processes; however, its genetic alterations could be implicated in the development and progression of diffused large B-cell lymphoma (DLBCL), Anaplastic Large Cell Lymphoma (ALCL), and Mantle cell lymphoma (MCL). DLBCL is the most common subtype of aggressive B-type lymphoma [90].

Several genetic alterations have been identified in DLBCL, and they lead to diverse approaches to classification, related to clinical course and response to therapy [91].

Genetic mutations involved in different pathways in DLBCL range widely among mechanisms related to B cell differentiation, B cell receptor signaling, activation of the NF-κB pathway, apoptosis and regulation of epigenetic [91].

As concerns PAX5, its role in DLBCL has been mainly associated with the occurrence of chromosomal translocations [92] t(9;14)(p13;q32) that place PAX5 under the control of the immunoglobulin heavy chain (IGH) locus enhancers, leading to its abnormal overexpression [93].

This aberrant activity can dysregulate gene networks, promoting lymphomagenesis by enhancing cellular proliferation or impeding normal differentiation [94]. PAX5 alterations associated with DLBCL include not only genetic, but also epigenetic changes.

Indeed, data from in vivo experiments found a correlation between DLCBL and PAX5 up-regulation induced by MTA1 [95]. Specifically, in DLBCL, overexpressed and activated MTA1 interacts with the p300 histone acetyltransferase, leading to its recruitment to the Pax5 promoter and the subsequent stimulation of Pax5 transcription [95]. Global gene profiling identified down-regulation of a set of genes, including those downstream of Pax5 and directly implicated in the B-cell lymphogenesis [95]. Moreover, PAX5 and its back-spliced circular RNA (circ1857) coordinate GINS1 expression in DLBCL cell line and xenografts [96]. This protein regulates both the initiation and progression of DNA replication and seems to be correlated with malignant transformation [97].

Among non-Hodgkin lymphomas, the mantle cell lymphoma (MCL) is characterized by dysregulation of cell cycle pathways, frequently driven by the t(11;14)(q13;q32) translocation, which results in CCND1 overexpression [98]. It is known that there are cyclin D1 negative mantle cell lymphomas characterized by an aggressive clinical evolution. This highlighted the need for a reliable biomarker to identify cyclin D1-negative mantle cell lymphomas, which tend to exhibit more aggressive clinical behavior. SOX 11, a nuclear protein expression, is a useful marker to identify mantle cell lymphoma, beyond cyclin D1 positivity [99].

While the central driver of MCL remains CCND1 de-regulation, alterations in PAX5 contribute to disease complexity by affecting B-cell identity, differentiation, and signaling. Indeed, MCL cells are derived from the mantle zone B cells which normally showed strong PAX5 expression, also clear in immunostaining [100].

MCL patients carrying a strong expression of PAX5 have been associated with a reduced overall survival, indicating PAX5 as an independent prognostic biomarker [101].

It was assumed that overexpression of SOX11 could further induce PAX5 expression [102].

This hypothesis could be explained by considering that most of the PAX5-related genes were positively correlated with SOX11/CyclinD1, boosting the activation of PI3K/AKT/mTOR and Wnt/β-catenin pathways [101].

Together with SOX11 and TP53 expression, PAX5 may enhance the stratification of MCL patients, as those with triple-positive expression exhibited inferior progression-free survival (PFS) and overall survival (OS), independent of the Mantle Cell Lymphoma International Prognostic Index (MIPI), Ki-67 levels, and TP53 mutation status [103].

It is interesting to report what is known about an aberrant PAX5 expression in lymphoid neoplasms with a non-B phenotype.

ALCL is considered a specific entity by the 5th World Health Organization (WHO) classification of hematolymphoid neoplasms. It originates from T cells [104] and is divided into the ALK-positive and ALK-negative subtypes.

As these are T-cell-derived neoplasms with monoclonal expression of the TCR, PAX5 is negative and is often used as a marker in differential diagnostics with other lymphomas, that morphologically overlap with ALCL, such as Hodgkin’s lymphoma, characterized by weak PAX5 expression [105].

However, there are known cases of ALK–ALCL showing aberrant PAX5 expression, which is also confirmed by the presence of extra copies [106].

This finding suggests that PAX overexpression contributes to lymphomagenesis [105,107,108,109].

Overall, PAX5 stands as versatile a contributor across lymphoma subtypes, particularly as an oncogenic driver in DLBCL and a prognostic marker in MCL. While PAX5 is often a secondary player in lymphomas, its genetic and epigenetic alterations are significant contributors to disease initiation, progression, and patient prognosis.

## 3. PAX5 and PAX6 in Breast Carcinomas

Breast cancer remains one of the most common and heterogeneous malignancies affecting women worldwide [110,111]. Despite advances in early detection and treatment, challenges persist in predicting disease progression and tailoring personalized therapies. In recent years, research has focused on identifying novel biomarkers that can improve diagnosis, prognosis, and therapeutic response [112,113,114,115]. Emerging molecular markers, including genetic, epigenetic, proteomic, and metabolomic signatures, offer promising insights into tumor behavior and patient outcomes. These new biomarkers have the potential to guide more precise treatment strategies, ultimately improving survival rates and quality of life for patients with breast cancer.

### 3.1. PAX5 as Tumor Suppressor in Breast Cancer

PAX5 has been linked to a broad spectrum of tumorigenic processes, extending beyond its canonical roles in leukemogenesis and lymphomagenesis. In breast cancer, numerous studies have demonstrated that PAX5 plays an inhibitory role in the invasive capacity of malignant cells.

For instance, it has been suggested that PAX5 could modulate the epithelial to mesenchymal transition (EMT) process in breast cancer cells by inducing pro-epithelial characteristics [116,117]. EMT is a biological process that enables cancer cells to invade surrounding tissues and form distant metastases [118,119,120,121].

In this context, PAX5 is considered a tumor suppressor gene [122]. The main mechanisms through which PAX5 is silenced in aggressive breast cancer concern the hypermethylation of its promoter [123,124]. This epigenetic modulation is generally regulated by a feedback loop, involving PAX5, miR-142, ZEB1 and DNMT1 [125]). Indeed, PAX5 promotes the expression of miR-142, which could affect breast cancer progression by regulating DNMT1 and ZEB, involved in the methylation of Pax5 promoter [125].

Breast cancer cells show increased methylation of PAX5 regulatory sequences, and ectopic induction promote increase expression of p53, p21, and p27, while CyclinD1, CDk4 and VEGF undergo down-regulation [126]. These data suggest the role of PAX5 in inducing cell cycle arrest and inhibiting cellular proliferation and invasion [126]. One of the main actors involved in these processes is the Focal Adhesion Kinase (FAK), a molecule involved in the regulation of cell adhesion, migration, proliferation and angiogenesis [127]. In breast cancer cell lines, PAX5 inversely correlates with pro-malignant FAK expression, thereby promoting the maintenance of epithelial cell characteristics [128]. The pro-epithelial feature associated with the activity of PAX5 has also been demonstrated by its active induction of E-cadherin expression [122]. E-cadherin is a member of adhesion molecules, crucial in maintaining the epithelial phenotype and considered a tumor suppressor [129]. Immunohistochemical evaluation found constitutive expression of PAX5 in mammary epithelial cells, promoting gene expression of E-Cadherin, while in breast cancer it suppressed breast cancer cell migration, invasion, and promoting cell adhesion properties [122]. PAX5 ability to reduce cells invasion in breast cancer cells has been proved also in MCF-7 and MDA-MB-231 cell lines [130]. Forced expression of PAX5 reduces gene expression of fibronectin and vimentin (mesenchymal molecules) and promotes the localization of β-catenin to the sites of cell–cell junctions. Notably, PAX5 is able to control the expression of non-coding-RNA, which could be equally associated with breast cancer progression. In biopsies derived from breast cancer patients, both PAX5 and miR-215 showed a reduced expression within the cancer lesions compared to adjacent normal tissue [116]. During malignant transformation as well as in the acquisition of invasive characteristics, PAX5 up-regulates other miRNA, such as microRNA-155 [131], a suppressor of malignant growth [132]. Conditional expression of microRNA-155 in breast cancer models highlighted its suppression of IKKε (IKBKE) expression, followed by down-regulation of NF-κB signaling, a process also modulated by PAX5 [131]. So, PAX5 inhibits NF-κB signaling and its downstream target IKKε through the regulation of microRNA-155. Although a greater number of scientific reports highlighted PAX5 as a tumor suppressor in breast cancer, in some cases it may have a dual face and act as promoter of the aggressive phenotype. For instance, in MCF-7 cells, resistance to Tamoxifen treatment could be linked to PAX5 [133]. Authors pointed out an activation of nuclear IKKα by cytokines which induce FAT10 expression, through PAX5 activity. This response confers Tamoxifen resistance in ER+ breast cancer cells [133].

Therefore, PAX5 plays a predominantly anti-proliferative role in breast cancer by modulating EMT, promoting pro-epithelial characteristics, and regulating tumor-suppressive pathways. However, the apparent dual role of PAX5 in breast cancer, as both a tumor suppressor and a potential promoter of aggressive phenotypes, may be attributed to the heterogeneity of breast cancer subtypes and the context-dependent nature of transcriptional regulation. It is possible that PAX5 plays distinct roles in luminal, HER2-positive, and triple-negative breast cancers, potentially modulating different target genes or interacting with distinct cofactors within each histological or molecular subtype.

Bioinformatic analysis performed using the UCSC Xena platform (https://xena.ucsc.edu/compare-tissue/), based on TCGA Breast Cancer (BRCA) cohort data (accessed on 28 March 2025), showed that PAX5 expression does not influence overall survival in breast cancer patients (Figure 1A). However, a significant increase in PAX5 expression was observed in both primary tumors and metastatic lesions compared to normal tissues (Figure 1B).

### 3.2. PAX6: A Double-Faced Actor in Breast Cancer

PAX6 plays an important and complex role in breast cancer. It is a transcription factor primarily known for its critical role in neuroectodermal tissue [134,135]. It regulates the differentiation and migration of neural progenitor cells, particularly in regions like the forebrain [136] and cerebral cortex [137]. In addition, PAX6 is involved in the differentiation and function of pancreatic islet cells, particularly α and β cells, through the regulation of hormones transcription such as insulin and glucagon [138].

Above all PAX6 is a key regulator of eye development, where it is essential for the formation of the lens, retina, cornea, and iris [139]. Indeed, due to its importance in the organogenesis of the eye, PAX6 heterozygous mutations are associated with aniridia [140], a neurodevelopmental disease which could lead to glaucoma. In most of these physiological regulatory mechanisms, PAX6 acts as an anti-apoptotic molecule both in the eye [141] and in adult neurons in the olfactory bulb [142], but also in pathological conditions such as retinoblastoma [143].

In the context of cancer, PAX6’s real nature is long way away from being understood, as it exhibits dual functionality, acting either as a tumor suppressor or a promoter of tumor progression depending on the molecular and cellular environment [144,145,146,147].

Regarding breast cancer progression, PAX6 also exhibits a dual mode of action. As a tumor suppressor, PAX6 has been associated with epigenetic modulation, such as methylation of its promoter [124,148]. The percentage of Pax6 promoter methylation was about 50% within the infiltrating ductal carcinomas cohorts [124,149], even if not always associated with a reduction in RNA expression. Methylation status of the Pax6 gene seems to be also affected by the breast cancer molecular feature. Indeed, a higher level of methylation within Pax6 promoter was found in ER/PR positive breast cancers compared to ER/PR negative one [150]. However, clinical investigation of infiltrating ductal carcinomas found a higher PAX6 protein expression in biopsies from ER-negative cases compared with ER-positive ones [151]. The mechanisms through which PAX6 may act as pro-tumorigenic molecule are several and induce an increased expression, especially in more aggressive tumors. Methylation on the CpG site in PAX6 gene endured a significant reduction during the progression from primary breast tumor to lymph node metastasis [152]. Up-regulation of PAX6 in aggressive breast cancer cells promotes migration and expression of MMP2 and MMP9 proteins, by directly binding their genes and thus activating their expression [153]. Its possible role as an oncogenic protein not only concerns its involvement in migration, but also in proliferation. Indeed, PAX6 expression could be directly modulated by p38, which is inhibited by TNFRSF9 (a member of TNF receptor members) [154]. Expression of TNFRSF9 in breast cancer is significantly decreased compared to adjacent normal counterpart, and the reduction increases in the breast cancer cells with metastatic and malignant features [154]. TNFRSF9 inhibits p38/MAPK phosphorylation, which in turn impairs PAX6 and leads to a decreased cellular proliferation [154]. Breast cancer cells knocked down for PAX6 arrest in the G0/G1-phase of the cell cycle and are unable to promote tumor growth [150]. When anti-apoptotic signals increase, the expression of PAX6 could pump the survival cascade. These signals could also derive from non-coding RNA. Indeed, in breast cancer cells, PAX6 expression may be favored by a long non-coding RNA named DANCR [155]. The different regulatory mechanism of PAX6 involving DANCR implies its modulation of miR-758-3p, allowing PAX6 expression and leading to a reduction in both apoptosis and autophagy in breast cancer cells [155].

PAX6 exhibits a multifaceted role in cancer progression, with its function intricately influenced by cellular context, epigenetic regulation, and non-coding RNA interactions. Its dualistic nature as both a tumor suppressor and promoter underscore the need for further research to unravel its potential as a biomarker and prognostic factor in breast cancer.

Analysis performed using the UCSC Xena platform (https://xena.ucsc.edu/compare-tissue/), based on data from the TCGA Breast Cancer (BRCA) cohort and accessed on 28 March 2025, revealed no significant differences in overall survival based on PAX6 expression levels (Figure 1C). Nonetheless, PAX6 expression was found to be significantly increased in primary breast tumors compared to normal breast tissues (Figure 1D).

## 4. PAX8 and Thyroid Carcinoma

PAX8 is essential in the thyroid gland embryogenesis. It is vital for the survival and proliferation of differentiated thyroid follicular cells and it controls thyroid hormone release [156]. PAX8 positivity is demonstrated in many cases of thyroid tumors in which it acts by regulating the proliferation rate of neoplastic cells [24,157,158].

One of the mechanisms associated with tumorigenesis of Follicular Thyroid Carcinoma (FTC) was identified the *PAX8*/*PPARγ* fusion oncogene. This fusion oncogene derives from a balanced translocation between chromosome 2 and 3, in which most of the coding sequence of *PAX8* (2q13) and the entire translated reading-frame of the gene of the liganded nuclear receptor-family member peroxisome proliferator activates receptor gamma (*PPARγ*) (3p25).

The new PAX8/PPARγ fusion protein (PPFP) is linked to the pathogenesis of FTC. The effect of this protein could relate to constitutive overexpression of the full-length *PPARγ* domain, interference with wild-type PPARγ function, alterations of PAX8 function, formation of novel fusion gene activities, or a combination of such events [159].

PPARγ regulates metabolic processes and plays a role in regulating cell cycle progression and apoptosis (cell death). The fusion protein has a dominant-negative effect on PPARγ-induced gene transcription in immortalized human thyroid cells. This, results in compromising normal differentiation and can promote abnormal cell growth [160].

In a study by Fan et al. [161], immunohistochemistry and Western blot analyses revealed that the expression levels of the PAX8-PPARγ fusion protein were significantly elevated in thyroid cancer tissues and cell lines compared to their normal counterparts. Furthermore, PAX8-PPARγ expression was found to correlate with key clinical parameters, including tumor differentiation, TNM stage, and lymph node metastasis, indicating a potential role in thyroid tumor development and progression.

To explore its functional relevance, thyroid cancer cell lines were treated with a PAX8-PPARγ-targeting antisense oligonucleotide (siRNA-PAX8-PPARγ), effectively silencing the fusion gene. This intervention resulted in a marked reduction in cancer cell proliferation, an increase in apoptosis, and decreased expression of the fusion protein.

Collectively, these findings support the hypothesis that PAX8-PPARγ contributes to thyroid cancer cell survival and proliferation, highlighting its potential as a therapeutic target.

A study conducted on papillary thyroid carcinoma (PTC) cells demonstrated that the long non-coding RNA PAX8-AS1, which targets PAX8, can suppress tumor malignancy. Overexpression of PAX8-AS1 in transfected PTC cells led to a significant reduction in cell proliferation and a concomitant increase in apoptosis. These results indicate that PAX8-AS1 may act as a tumor suppressor, impairing the proliferative capacity and enhancing the apoptotic response of PTC cells [162].

Bioinformatic analysis performed using the UCSC Xena platform (https://xena.ucsc.edu/compare-tissue/), based on data from the TCGA Thyroid Cancer (THCA) cohort and accessed on 28 March 2025, revealed that PAX8 expression does not significantly influence overall survival in thyroid carcinoma patients (Figure 2A). However, a significant decrease in PAX8 expression was observed in both primary tumors and metastatic lesions compared to normal thyroid tissues, suggesting a potential role for PAX8 down-regulation in thyroid tumorigenesis (Figure 2B).

## 5. PAX2 and PAX8 in Tumors of the Urogenital Tract

### Renal Cell Carcinoma

PAX2 and PAX8 play essential roles in the early stages of renal development, influencing nephron formation, ureteric bud development, and the morphogenesis of the entire urinary system [23,163,164].

PAX2 is one of the most crucial transcription factors in renal development, tightly regulated during normal kidney development and down-regulated when the organ is full grown.

It promotes the mesenchymal-to-epithelium conversion, leading their differentiation into renal tubular structures [165].

PAX8, in a human adult kidney, is still expressed in the epithelial cells, as well as in areas including renal stem cells, particularly in the parietal cells of the Bowman’s capsule and in the medullary regions [166].

In the context of renal carcinoma, the role of PAX proteins is of growing interest, particularly in relation to the pathogenesis and progression of renal cell carcinoma (RCC), the most common type of urogenital cancer [167].

Both PAX2 and PAX8 can modulate similar cellular pathways, such as proliferation, differentiation, survival, metastasis and metabolism in a cancer context.

The role of PAX8 in the context of renal cell carcinoma is linked to angiogenesis, which supports tumor growth even under hypoxic conditions. One of the key features of clear cell RCC is the presence of hypoxic conditions within the tumor microenvironment due to aberrant VHL (von Hippel-Lindau) pathway activation, which stabilizes HIF-1α (hypoxia-inducible factor 1 alpha) and promotes angiogenesis [168,169].

An additional mechanism recently linked to PAX8 in RCC involves the regulation of cell proliferation. Specifically, silencing PAX8 expression results in a significant reduction in the proliferative capacity of RCC cells, highlighting its potential role in tumor growth and maintenance [170].

Bioinformatic and single-cell transcriptomic analyses were performed to investigate PAX8 expression in clear cell carcinomas (Figure 3). Based on data from the TCGA Kidney Clear Cell Carcinoma (KIRC) cohort, accessed through the UCSC Xena platform on 28 March 2025, patients with higher PAX8 expression showed reduced overall survival compared to those with lower expression levels (Figure 3A). Additionally, a significant decrease in PAX8 expression was observed in primary tumors relative to normal kidney tissues (Figure 3B). Mutational analysis revealed that alterations in the long non-coding RNA PAX8-AS1 were significantly associated with a poorer prognosis, as patients harboring these mutations exhibited reduced overall survival (Figure 3C).

Single-cell transcriptomic data, accessed via the TISCH portal on 28 March 2025, showed that PAX8 is highly expressed in multiple cell populations, including cancer cells (Figure 3E). Furthermore, cell-type-specific analysis demonstrated that malignant cells display higher levels of PAX8 expression compared to inflammatory cells, highlighting its potential role in tumor cell identity and function (Figure 3E).

PAX2 regulates genes involved in cell adhesion and the extracellular matrix (ECM), to promote tumor cell invasion and metastasis. In a study form Doberstein et al., PAX2 can regulate the expression of a metalloprotease (ADAM-10) in renal cancer cells. The down-regulation of ADAM10 lead to the loss of E-cadherin, which is observed during EMT [171].

There is also evidence of a direct relationship between TGF-β1 signaling and *PAX2* expression during RCC tumor progression. TGF-β1 signaling in CC-RCC cells results in the direct inhibition of *PAX2* expression through SMAD-mediated transcriptional suppression of the *PAX2* gene promoter. These data therefore provide a clearer understanding of the role of TGF-β1 signaling and *PAX2* expression in cancer [172].

## 6. PAX8 and PAX9 in the Tumors of the Lung

Lung cancer is primarily divided into two major categories: Non-Small Cell Lung Cancer (NSCLC), accounting for 85–90% of lung cancers, and Small Cell Lung Cancer (SCLC), characterized by rapid growth and early metastasis [173]. NSLCL comprises adenocarcinoma, frequently characterized by EGFR, ALK, or KRAS mutations; squamous cell carcinoma, associated with alterations in TP53 and FGFR1; and large cell carcinoma, aggressive and often diagnosed at advanced stages [174].

Regarding PAX8 and PAX9, recent scientific studies have linked these transcription factor to lung cancer, despite their expression not being required in the lung under physiological conditions [175].

In lung carcinoma, while the oncogenic role of PAX8 is yet to be explored, it is crucial in the diagnostic workup for distinguishing primary tumors from metastases, especially in cases resembling other PAX8-positive cancers (e.g., thyroid or renal cancers) [176,177].

In mediastinal mass biopsies, PAX8 could be useful to discriminate the origin of cancer cells, because its specificity in distinguishing thymic carcinoma from carcinomas originating from the lung [178,179]. For instance, thymic carcinoma cells more strongly express PAX8, CD5, and CD117 compared to lung cancer cells.

Moreover, lung very often represents a target for metastasizing for highly expressing PAX8 tumors, such as renal [176], thyroid [177], and ovarian [180] tumors.

As for other PAX proteins, PAX9 plays a critical role in the development of various tissues and organs during embryogenesis. They pivot regulatory function of PAX9 is associated with odontogenesis and craniofacial development [181,182]. PAX9 is vital for the formation of teeth, particularly in initiating the tooth development process and dental epithelial–mesenchymal interactions [183]. It also takes part in development of pharynges and skeletal apparatus [184] and regulates the formation of the thymus [185], parathyroid glands, and the esophagus [184].

Concerning the oncogenic role of PAX9, an up-regulation of mRNA and protein levels have been detected in lung adenocarcinoma (LUAD) tissues and cell lines (A549 and H-1299) compared to healthy counterparts [186]. In LUAD patients, higher expression of PAX9 is associated with poorer prognosis and enhanced tumor progression [187], while in cell lines it regulates significantly reduces cell proliferation, migration, and invasion, suggesting its role in promoting aggressive cancer behavior [186]. The promotion of cancer development is confirmed also in SCLC, where PAX9 acts as a key downstream target of the BAP1/ASXL3/BRD4 chromatin remodeling complex [188].

Mechanistically, PAX9 interacts with components of the nucleosome remodeling and deacetylase (NuRD) complex to block expression of tumor suppressor genes [188]. Genetic depletion or pharmacological inhibition of PAX9 can reactivate these enhancers, promoting tumor-suppressive pathways and neural differentiation. From a genomic localization, Pax9 gene sits within Chr 14q13.3, a genomic region which frequently undergoes amplification in lung cancer [189]. Gene mapping revealed three genes (TTF1/NKX2-1, NKX2-8, PAX9) in the core region, all of which encode transcription factors involved in lung development [190]. Continuous expression of NKX2-8 and PAX9 may be essential to the tumor maintenance of amplified SCC cells, and both gene knockdown and overexpression experiments further supported oncogenic roles for these genes. Overall, PAX9 influences both the cellular behavior and epigenetic landscape of lung cancer, highlighting its dual role as a biomarker and a potential therapeutic target in lung cancer subtypes.

Therefore, PAX8 and PAX9 play distinct roles in lung cancer biology, diagnostics, and potential therapeutic applications. On one hand, PAX8 acts primarily as a diagnostic marker to differentiate lung cancers from metastases originating in PAX8-positive tissues (e.g., thyroid, renal, ovarian cancers); on the other hand, PAX9 demonstrates a more direct role in lung cancer progression, particularly in lung adenocarcinoma (LUAD) and small cell lung cancer (SCLC), where it may help in prognostic determination.

## 7. PAX2 and PAX8 in Ovarian Cancer

PAX2 and PAX8 also play important roles in the development of female reproductive organs. Both PAX2 and PAX8 are critical regulators of Müllerian duct-derived tissues (such as the ovaries, fallopian tubes, and endometrium). PAX8 is crucial for maintaining the function of the epithelial cells of the ovaries, and its expression is tightly regulated during development [191].

Ovarian cancer is the fifth leading cause of cancer-related death in women and the most lethal gynecological malignancy. High-grade serous carcinoma (HGSC) accounts for 80% of ovarian cancer cases and it is the deadliest histological subtype of epithelial ovarian cancer (EOC) [192,193].

PAX8 is frequently expressed in serous, endometrioid, and clear cell ovarian carcinomas, which are thought to arise from the Müllerian epithelium [194].

In HGSC, PAX2 and PAX8 can enhance tumor vascularization via their interaction with SOX17, another developmental factor. They support the rapid growth of the tumor and its ability to spread to other organs through the blood or the lymphatic system. This angiogenic activity is important for the metastatic potential of HGSC. The PAX8 and SOX17 complex in ovarian cancer cells promoted the secretion of angiogenic factors by suppressing the expression of *SERPINE1*, which encodes a proteinase inhibitor with anti-angiogenic effects [195].

Moreover, PAX8 promotes proliferation and has an anti-apoptotic role in ovarian cancer cells. The protein positively regulates the expression of TP53 and p21; the latter localizes to the cytoplasm of HGSC cells where it plays a non-canonical, pro-proliferative role [196]. Kakun et al. have published on the pro-proliferative and anti-apoptotic role of PAX8, via transcriptional activation of p21, also in uterine serous papillary carcinoma [197].

By regulating genes involved in cell cycle control, PAX8 can facilitate tumor growth in HGSC. The silencing of PAX8 in cancer cells induces cell-cycle arrest and leads to reduction in the expression of E2F1 and its target genes [198].

With regard to PAX2, it seems to be involved in the promotion of fatty acid metabolic reprogramming, a shift from normal oxidative metabolism to lipid biosynthesis pathways that support rapid cancer cell growth and survival. PAX2 regulates the expression of key enzymes in fatty acid synthesis, such as FASN (Fatty Acid Synthase), which is commonly overexpressed in cancers. This high expression of key enzymes in fatty acid metabolism was associated with a shorter progression-free survival time in patients with serous ovarian cancer [199].

On the other hand, PAX2 expression could be involved in maintaining the epithelial characteristics of the tumor cells, in Low Malignant Potential (LMP) ovarian tumors. This might explain a role in the limited malignancy of these tumors.

PAX2 expression is stronger in LMP tumors and lower in more aggressive cancer types like HGSC. In LGSC, PAX2 might serve as a prognostic marker. The higher expression levels possibly correlate with a better prognosis due to the tumor’s less aggressive behavior and the loss of PAX2 expression might be linked to a more aggressive or invasive phenotypes [200].

In conclusion, understanding PAX2’s and PAX8’s distinct roles in ovarian tumorigenesis helps refine diagnostic and prognostic strategies and offers potential avenues for developing targeted therapies to treat ovarian cancer more effectively.

## 8. Other PAX’s

PAX3 plays a crucial role in embryonic development, particularly in myogenesis and neural crest cell differentiation, and is implicated in tumorigenesis of rhabdomyosarcoma, glioma, and melanoma through gene fusions or overexpression [201,202,203]. It promotes proliferation, inhibits apoptosis, and activates oncogenic pathways, positioning it as a potential prognostic biomarker and therapeutic target.

PAX1, by contrast, acts mainly as a tumor suppressor, especially in cervical cancer, where promoter hypermethylation silences its expression [204]. Its methylation status serves as a diagnostic and prognostic biomarker and may influence radiotherapy response, making it a candidate for precision oncology strategies [10,204].

Table 2 summarizes the main molecular mechanisms associated with PAX expression in human cancers.

## 9. Diagnostic Relevance and Clinical Applications of PAX Family Proteins

Group II PAX members are commonly used as additional immunohistochemical markers for cancer diagnosis. For instance, PAX5 is employed as an immunohistochemical marker for the diagnosis and subtyping of lymphomas. PAX5 expression is absent in anaplastic large cell lymphomas, therefore PAX5 positivity in Hodgkin lymphoma cells can be used to differentiate Hodgkin’s lymphoma from anaplastic large cell lymphoma [105].

PAX2 and PAX8 are valuable markers in the differential diagnosis of neoplasms, particularly in poorly differentiated tumors or metastatic lesions of unknown primary origin. The expression of transcriptional programs is essential for maintaining the identity of the cell of origin under normal physiological conditions. While this expression can become deregulated during tumorigenesis, it remains a valuable diagnostic tool.

In non-neoplastic tissue, PAX8 exhibits strong nuclear staining in follicular cells of the thyroid, Müllerian epithelial cells, and renal tubular epithelium. In benign tumors and precursor lesions, PAX8 expression is confirmed in thyroid follicular adenomas and renal oncocytomas, both of which demonstrate strong and diffuse immunoreactivity. Furthermore, strong nuclear positivity for PAX8 is maintained in carcinomas originating from the thyroid, kidney, and Müllerian tract [197].

The persistent expression of PAX8 and PAX2 in tumors originating from renal cells emphasizes their importance in the diagnostic workup of both primary and metastatic renal tumors. When combined with other markers, such as CD10, RCC markers, and cytokeratin, PAX2 and PAX8 assist pathologists in accurately identifying RCC and its subtypes, thereby aiding in appropriate treatment planning [214,215].

PAX2 is particularly useful for diagnosing specific RCC subtypes, such as papillary RCC and chromophobe RCC, which retain more embryonic renal features. Its expression is crucial in confirming a renal origin for the tumor of known origin. PAX8, on the other hand, is most used in diagnosing RCC, although it can also be valuable in identifying papillary RCC.

The widespread expression of PAX2 and PAX8 positions them as promising candidates for further research into their potential as therapeutic targets [216]. In non-neoplastic tissue, PAX8 shows strong nuclear staining in thyroid follicular cells, Müllerian epithelial cells, and renal tubular epithelium. Regarding benign tumors and precursor lesions, PAX8 expression is confirmed in thyroid follicular adenomas and renal oncocytomas, both of which show strong and diffuse immunoreactivity. Additionally, a strong nuclear positivity for PAX8 is maintained in carcinomas from the thyroid, kidney, and Müllerian tract [217]. In mediastinal mass biopsies, PAX8 is crucial in the diagnostic workup for distinguishing primary tumors from metastases, particularly in cases that resemble other PAX8-positive cancers (e.g., thyroid or renal cancers). It can help discriminate the origin of cancer cells due to its specificity in distinguishing thymic carcinoma from poorly differentiated lung carcinoma [178].

Thymic carcinoma cells, for instance, more strongly express PAX8, CD5, and CD117 compared to lung cancer cells [178,179].

PAX8 may also serve as a useful marker in the differential diagnosis of other neoplasms with mediastinal localization. For instance, thyroid carcinoma, esophageal carcinoma, and germ cell tumors express PAX8.

Thus, PAX8 exemplifies how a single gene can play a multifaceted role in cancer biology, its dysregulation contributing to the onset and progression of specific carcinomas, while simultaneously serving as a valuable diagnostic and prognostic biomarker and representing a promising target for therapeutic intervention [194].

## 10. Conclusions

Numerous studies acknowledge the importance of PAX genes in both the embryological development of various systems and in malignant neoplastic pathology. In particular, the key role of these transcription factors places them at the heart of critical oncogenic pathways, functioning as both promoters and suppressors. Even in cases where they are not the primary drivers, they can influence the activity of signaling involved in cancer progression such as EMT and cell death. Moreover, the expression of PAX genes appears to be tissue-specific, spanning from certain normal tissues to the tumors that arise from them. A recent review by Shaw et al., 2024 [204], has addressed the general biological functions and regulatory mechanisms of the PAX gene family, with emphasis on their roles in development and disease. While that work provides a valuable overview of PAX protein interactions and signaling pathways, our review aims to offer a cancer-focused, clinically oriented analysis. Specifically, we explore the role of PAX genes in selected malignancies, where their oncogenic and tumor-suppressive functions are best characterized. We detail genetic alterations, fusion proteins, and epigenetic modulations affecting PAX activity, and discuss their implications for diagnosis, prognosis, and potential targeted therapies. Furthermore, we integrate publicly available omics datasets (e.g., TCGA, UCSC Xena) to support our analysis with survival data and tumor-specific expression profiles. In this way, our review complements existing literature by providing a comprehensive synthesis tailored to translational oncology and diagnostic pathology.

According to these findings, future studies based on multidisciplinary investigations, including anatomic pathology, diagnostic imaging, and multiomic approaches [218,219,220,221,222], should aim to further explore the potential of PAX molecules as therapeutic targets and prognostic factors.

## Figures and Tables

**Figure 1 diagnostics-15-01420-f001:**
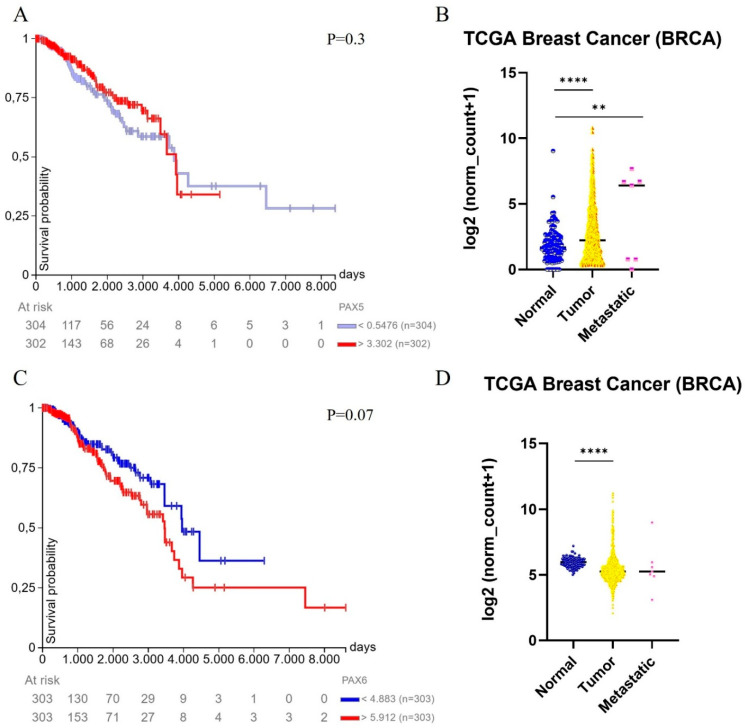
Bioinformatic analysis of PAX5 and PAX5 expression in Breast Carcinomas. (**A**) The expression of PAX5 does not influence overall survival in breast cancer patients. (**B**) A significant increase in PAX5 expression is observed in primary tumors and metastatic lesions compared to normal tissues. (**C**) Overall survival of breast cancer patients based on the expression of PAX6. (**D**) A significant increase in PAX6 expression is observed in primary tumors compared to normal tissues. Gene expression data from the TCGA Breast Cancer (BRCA) cohort are available through the UCSC Xena platform (https://xena.ucsc.edu/compare-tissue/), accessed on 28 March 2025. ** *p* < 0.01, **** *p* < 0.0001.

**Figure 2 diagnostics-15-01420-f002:**
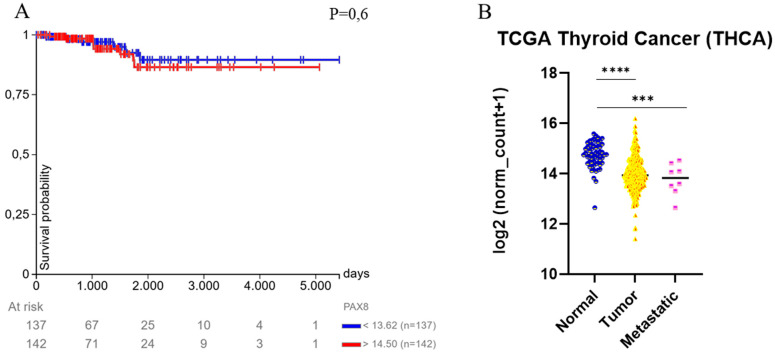
Bioinformatic analysis of PAX8 expression in a thyroid carcinoma. (**A**) The expression of PAX8 does not influence overall survival in thyroid cancer patients. (**B**) A significant decrease in PAX8 expression is observed in primary tumors and metastatic lesions compared to normal tissues. Gene expression data from the TCGA Thyroid Cancer (THCA) cohort are available through the UCSC Xena platform (https://xena.ucsc.edu/compare-tissue/), accessed on 28 March 2025. *** *p* < 0.001; **** *p* < 0.0001.

**Figure 3 diagnostics-15-01420-f003:**
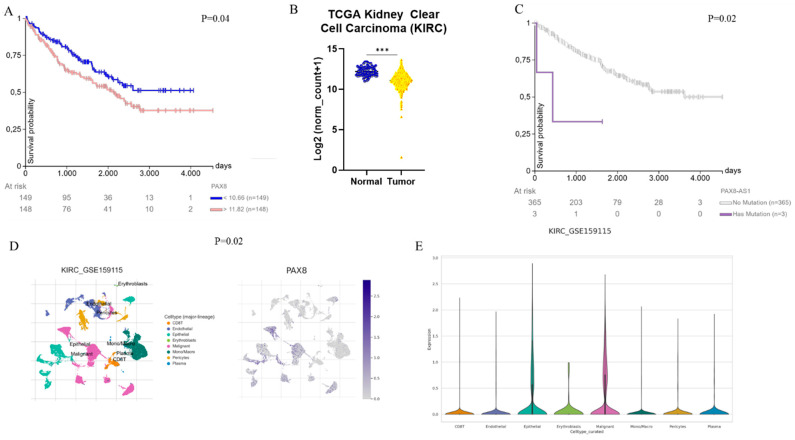
Bioinformatic and single cell transcriptomic analysis of PAX8 expression in clear cell carcinomas. (**A**) Patients with clear cell carcinoma exhibit reduced overall survival compared to those with lower expression levels. (**B**) A significant decrease in PAX8 expression is observed in primary tumors as compared to normal tissues. (**C**) Mutations in PAX8-AS1 are significantly associated with reduced overall survival, as patients harboring these mutations exhibit poorer prognoses. (**D**) Single-cell transcriptomic analysis reveals high expression of PAX8 in several cell types, including cancer cells. (**E**) Cell-type-specific analysis shows higher PAX8 expression in malignant cells compared to inflammatory cells. Gene expression data from the TCGA Kidney Clear Cell Carcinoma (KIRK) cohort are available through the UCSC Xena platform (https://xena.ucsc.edu/compare-tissue/), accessed on 28 March 2025 (Panel **A**–**C**). Single-cell transcriptomic data for PAX8 are available through the TISCH portal (http://tisch.comp-genomics.org/), accessed on 28 March 2025. *** *p* < 0.001.

**Table 1 diagnostics-15-01420-t001:** Pathophysiological role of PAX proteins in humans.

Group I	Expression During Development	Expression in Adults	Role in Tumorigenesis
**PAX1**	Human skeleton, parathyroid glands and thymus [6,7]	Low levels in esophagus, skeletal muscle, kidneys, pituitary gland, skin and thyroid [8,9,10]	Tumor suppressor role in various cancers (e.g., cervical cancer, ovarian cancer, colorectal cancer). Highly methylated in these tumors [8,9,10]
**PAX9**	Human skeleton, parathyroid glands and thymus [7]	PAX9 expression is restricted to the endocrine tissues, lymphatic system, cervix, bronchus, tongue, esophagus and salivary gland [11]	Genetic alterations contribute to carcinogenesis, common in lung adenocarcinoma and squamous cell carcinoma [12]
**Group II**	**Expression during development**	**Expression in adults**	**Role in tumorigenesis**
**PAX2**	Central nervous system, optic vesicle, optic disk, optic nerve, ears, kidneys, pancreas, female reproductive tract (cervix, fallopian tubes, uterus) and adult testis [13,14]	Low expression in the brain, pancreas, pituitary gland, testis, uterus; moderate levels in cervix, fallopian tubes and kidneys [15]	Highly expressed in primary renal carcinomas. Inhibition induces rapid apoptosis in bladder carcinoma cell lines. Important for growth and survival of several urogenital cancers [16]
**PAX5**	Involved in the differentiation of hematopoietic stem cells into mature B cells [17]	Expressed in adult human brain, spleen, colon and testis [18,19,20]	A potent oncogene in hematological cancers, particularly lymphoma and lymphocytic leukemia. Down-regulation blocks B-cell differentiation, giving cells the ability to proliferate, evade, and resist apoptosis [21,22]
**PAX8**	Necessary for proper development of thyroid, testis, kidneys and fallopian tubes [23,24]	Some isoforms expressed in adult tissues [23,24]	Expressed in most thyroid cancers, correlating with higher risk of tumor recurrence and plays a role in progression of Follicular Thyroid Carcinoma [25,26]
**Group III**	**Expression during development**	**Expression in adults**	**Role in tumorigenesis**
**PAX3**	Important regulator of neural tube development, neural crest formation, skeletal muscle development [27]	Expressed in adult human adipose tissue, arterial tissue, brain, breast, cervix, minor salivary gland, skeletal muscle, prostate, skin, and testis [28]	Mutations associated with Waardenburg syndrome (types I and III), embryonal and alveolar rhabdomyosarcoma [29,30]
**PAX7**	Important regulator of neural tube development, neural crest formation, skeletal muscle development [27]	Satellite cells in skeletal muscle [31,32]	Mutations associated with Waardenburg syndrome (types I and III), embryonal and alveolar rhabdomyosarcoma [29,30]
**Group IV**	**Expression during development**	**Expression in adults**	**Role in tumorigenesis**
**PAX4**	Plays a fundamental role in the development of pancreas and gastrointestinal tract cells [33]	Colon and small intestine [34]	Dysregulation linked to developmental disorders, diabetes, and tumors of the exocrine pancreas and intestine [35,36]
**PAX6**	Important for the development of eyes, brain, and pituitary gland	Brain, kidneys, pituitary gland, pancreas, testis [37]	Dysregulation linked to developmental disorders, diabetes, and tumors of the exocrine pancreas and intestine [35,36]

**Table 2 diagnostics-15-01420-t002:** Main molecular mechanisms of PAX proteins in cancer.

Molecular Mechanism	Description	Examples
**Regulation of Cell Survival and Proliferation**	PAX proteins can inhibit p53 expression, influencing cell survival and proliferation	**PAX2, PAX5, PAX8** (human astrocytoma): Inhibit p53 expression by direct DNA binding in the 5′ region of the p53 gene [205] **PAX3**: Induces apoptosis in rhabdomyosarcoma cells by targeting PAX3 mRNA translation start site ([206]) **PAX2** (rat renal proximal tubular cells): Stimulates expression in response to Angiotensin II and affects apoptosis vs. proliferation [207]
**Regulation of Cell Fate and Differentiation**	Many PAX proteins influence specific cell lineage differentiation and maturation.	**PAX5**: Regulates B-cell maturation; pro-B cells lacking Pax5 fail to differentiate without Pax5 restoration [21] **PAX6**: Inactivation at retinal progenitor stage leads to loss of pluripotency [208].
**Regulation of Invasion and Metastasis**	PAX proteins can promote invasion and metastasis by regulating cell adhesion and migration processes.	**PAX2** (renal cell carcinoma): Promotes cell migration by upregulating matrix metalloproteinases (MMPs), facilitating cancer cell invasion [209]. **PAX8** (hepatocellular carcinoma): Regulates cell adhesion, migration, and invasion: enhanced expression of cyclin D1, vimentin, N-cadherin; on the contrary, a decreased expression of E-cadherin [210].
**Oncogenic Fusion Proteins**	PAX proteins can form fusion proteins due to chromosomal translocations, leading to aberrant activation of oncogenic pathways.	**PAX3-FOXO1 fusion protein** arises from a chromosomal translocation between the PAX3 gene on chromosome 2 and the FOXO1 gene (also known as FKHR) on chromosome 13. The resulting fusion protein has profound effects on the development and progression of rhabdomyosarcoma prevents myogenic differentiation, maintaining cells in an undifferentiated, proliferative state [211]. **PAX7**: Forms fusion proteins in alveolar rhabdomyosarcoma, promoting cell cycle progression and inhibiting apoptosis [212]. **PAX8-PPARγ fusion protein** (follicular thyroid carcinoma): Alters gene expression, blocks differentiation, and promotes tumorigenesis [213].

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
