# Peer review of "PAX Family, Master Regulator in Cancer"

_diagnostics, 2025, doi:10.3390/diagnostics15111420_

Round 1
Reviewer 1 Report
Comments and Suggestions for Authors
The review manuscript summarized the function of PAX protein. Although multiple PAX proteins have been discussed in both physiological and pathological conditions, the manuscript could be organized in a clearer way by rearranging the contents, adding more focus on diagnosis, and organize the storyline by disease, function, or by PAX protein types.
1. There’s a review article published last year on the same topic: ( https://doi.org/10.3390/cancers16051022), which also covers the interaction and regulation mechanisms of PAX family protein, and their roles in development, different diseases, and cancer. Please highlight the difference between the previous paper and this manuscript.
2. Table 1 showed that PAX7 is not expressed in adult cells; However, PAX7 is expressed in adult skeletal muscle as a crucial transcription factor expressed in quiescent and newly activated satellite cells, which are essential for muscle regeneration. PAX7 is important in maintaining the satellite cell pool and regulating its regenerative capacity. Please update with support from the literature.
3. The manuscript is organized by a mix of function and protein, which is a bit confusing and hard to follow. The main body started with around 6 pages (1/3 of the entire manuscript) of discussion of PAX5’s different roles across B cell development, hematological malignancies, as well as solid tumor like bread cancer. Then the manuscript started to group the sections by cancer types, such as PAX5 and 6 in breast cancer, PAX8 in thyroid cancer, and then PAX2 and 8 in renal cancer, and PAX8 and 9 in lung. The discussion is not particularly helpful for readers to follow and identify useful information. Additionally, ovarian cancer is typically not considered as a urogenital cancer, although the ovary belongs to the female reproductive system.
4. The author should provide more implications on clinical diagnosis, e.g., will the expression of PAX proteins support the diagnosis of certain diseases? Although the authors discussed the expression of PAX2 and PAX8 in diagnosis, but it’s under the section of “PAX2 and PAX8 in tumors of the urogenital tract”, which was not the most accurate title to capture the content. The manuscript should have a separate dedicated section to discuss the diagnosis and potential applications in the future. Listing the prognosis and expression level, which was how most of the plots are generated, is not entirely sufficient to support the application of PAX proteins in diagnosis; more solid research that demonstrates the specificity of those meachanisms are critical.
5. In order to be used as biomarkers, the proteins need to be specific and have distinct expression profiles between disease / normal cells. Therefore, in table1, please put more effort in adding how these PAX proteins are expressed in normal adult tissue and how its expression is different in tumor cells. Please also provide literature (which was missing in the majority of the cells in the middle column)
Author Response
REVIEWER#1
The review manuscript summarized the function of PAX protein. Although multiple PAX proteins have been discussed in both physiological and pathological conditions, the manuscript could be organized in a clearer way by rearranging the contents, adding more focus on diagnosis, and organize the storyline by disease, function, or by PAX protein types.
Reply: We sincerely thank the reviewer for the encouraging and thoughtful feedback.
- There’s a review article published last year on the same topic: ( https://doi.org/10.3390/cancers16051022), which also covers the interaction and regulation mechanisms of PAX family protein, and their roles in development, different diseases, and cancer. Please highlight the difference between the previous paper and this manuscript.
Reply: We thank the reviewer for highlighting the recent review article (Cancers 2024, 16(5), 1022), which indeed provides an important overview of the regulatory mechanisms and developmental roles of the PAX family proteins. However, our current manuscript offers a distinct and complementary contribution to the field. While the previous article focuses primarily on the molecular biology and developmental functions of PAX genes, our review is structured around the clinical relevance of PAX proteins in oncology, with in-depth discussion of their roles in specific malignancies including B-cell acute lymphoblastic leukemia, breast, thyroid, renal, and ovarian cancers. We place particular emphasis on genetic alterations (e.g., point mutations, deletions, translocations), the formation of oncogenic fusion proteins, and epigenetic modifications, integrating data from bioinformatics platforms such as TCGA and UCSC Xena. Moreover, we highlight the diagnostic and prognostic utility of PAX expression in human cancers and present it in a format tailored to oncologists, pathologists, and translational researchers. We believe this clinically-oriented, cancer-focused perspective, supported by current omics data, distinguishes our review from the broader, developmentally centered work published previously.
In the new version of our paper we added the following conclusions:
Numerous studies acknowledge the importance of PAX genes in both the embry-ological development of various systems and in malignant neoplastic pathology. In particular, the key role of these transcription factors places them at the heart of critical oncogenic pathways, functioning as both promoters and suppressors. Even in cases where they are not the primary drivers, they can influence the activity of signaling in-volved in cancer progression such as EMT and cell death. Moreover, the expression of PAX genes appears to be tissue-specific, spanning from certain normal tissues to the tumors that arise from them. A recent review by Shawet al., 2024 [204] has addressed the general biological functions and regulatory mechanisms of the PAX gene family, with emphasis on their roles in development and disease. While that work provides a valuable overview of PAX protein interactions and signaling pathways, our review aims to offer a cancer-focused, clinically oriented analysis. Specifically, we explore the role of PAX genes in selected malignancies where their oncogenic and tumor-suppressive functions are best characterized. We detail genetic alterations, fusion proteins, and epigenetic modulations affecting PAX activity, and discuss their implications for diagnosis, prog-nosis, and potential targeted therapies. Furthermore, we integrate publicly available omics datasets (e.g., TCGA, UCSC Xena) to support our analysis with survival data and tumor-specific expression profiles. In this way, our review complements existing liter-ature by providing a comprehensive synthesis tailored to translational oncology and diagnostic pathology.
- Table 1 showed that PAX7 is not expressed in adult cells; However, PAX7 is expressed in adult skeletal muscle as a crucial transcription factor expressed in quiescent and newly activated satellite cells, which are essential for muscle regeneration. PAX7 is important in maintaining the satellite cell pool and regulating its regenerative capacity. Please update with support from the literature.
Reply: thanks for this point out. We modified the table 1 according to the reviewer comment.
- The manuscript is organized by a mix of function and protein, which is a bit confusing and hard to follow. The main body started with around 6 pages (1/3 of the entire manuscript) of discussion of PAX5’s different roles across B cell development, hematological malignancies, as well as solid tumor like bread cancer. Then the manuscript started to group the sections by cancer types, such as PAX5 and 6 in breast cancer, PAX8 in thyroid cancer, and then PAX2 and 8 in renal cancer, and PAX8 and 9 in lung. The discussion is not particularly helpful for readers to follow and identify useful information. Additionally, ovarian cancer is typically not considered as a urogenital cancer, although the ovary belongs to the female reproductive system.
Reply: To differentiate our work from the previously published review in Cancers (2024, 16(5), 1022), which explored regulatory mechanisms and interactions of the PAX family across development and disease contexts, we deliberately chose to organize our manuscript around the role of specific PAX members in distinct neoplasms. This structure was intended to provide a more clinically oriented and oncology-focused perspective, emphasizing the diagnostic, prognostic, and therapeutic implications of individual PAX proteins in defined cancer types. This organization also allowed us to integrate original bioinformatic analyses, including expression patterns and survival data from public datasets (e.g., TCGA), which were not included in the previous review. We thank the reviewer for the valuable observation regarding ovarian cancer; accordingly, we have removed it from the section on urogenital tumors and placed it in a dedicated chapter on malignancies of the female reproductive system.
- The author should provide more implications on clinical diagnosis, e.g., will the expression of PAX proteins support the diagnosis of certain diseases? Although the authors discussed the expression of PAX2 and PAX8 in diagnosis, but it’s under the section of “PAX2 and PAX8 in tumors of the urogenital tract”, which was not the most accurate title to capture the content. The manuscript should have a separate dedicated section to discuss the diagnosis and potential applications in the future. Listing the prognosis and expression level, which was how most of the plots are generated, is not entirely sufficient to support the application of PAX proteins in diagnosis; more solid research that demonstrates the specificity of those meachanisms are critical.
Reply: We thank the reviewer for this insightful comment. In response, we have added a dedicated section titled “Diagnostic Relevance and Clinical Applications of PAX Family Proteins” to address more directly the role of PAX proteins in clinical diagnosis. In this section, we discuss the validated diagnostic utility of PAX2, PAX5, and PAX8 in specific cancer types, particularly their application in immunohistochemistry to support lineage identification and tumor classification, such as PAX8 in thyroid, renal, and Müllerian tumors, and PAX5 in B-cell lymphomas and leukemias. We also highlight examples where PAX fusion proteins or mutations (e.g., PAX5/ETV6 in B-ALL, PAX8/PPARγ in follicular thyroid carcinoma) serve as molecular diagnostic markers. This addition complements the bioinformatic survival and expression data presented earlier in the manuscript and strengthens the clinical translational value of our review. Furthermore, we agree with the reviewer that the previous title "PAX2 and PAX8 in tumors of the urogenital tract" did not fully reflect the diagnostic focus of the section. We have revised the manuscript structure accordingly to improve clarity and alignment with clinical applications.
- In order to be used as biomarkers, the proteins need to be specific and have distinct expression profiles between disease / normal cells. Therefore, in table1, please put more effort in adding how these PAX proteins are expressed in normal adult tissue and how its expression is different in tumor cells. Please also provide literature (which was missing in the majority of the cells in the middle column)
Reply: thanks for this point pout. In the new version of our manuscript we deeply modified the table 1 according to the reviewer’s suggestions.
Reviewer 2 Report
Comments and Suggestions for Authors
The role of genomic changes and expressed proteins in carcinogenesis and their prognostic importance are the subjects of many studies. I congratulate the authors for their article that generally addresses the PAX family in all cancers.
The article is generally well-written, written in a language that provides a pleasure to read and contains scientific richness. I believe that it will contribute to the literature. My general views and suggestions regarding the study are as follows.
Figures and tables are used appropriately and explanatorily throughout the study.
When mentioning the subgroups of PAX genes and proteins, the types of subgroups are stated in parentheses more than once, I think it would be sufficient to state it once.
In the first section, the functions of PAX genes and proteins, their subgroups are explained appropriately and supported by a table.
Was ‘II.1’ used incorrectly in the expression II.1 PAX5 (paired box 5), used on line 87? Check please.
In the second section, the role and mechanism of PAX-5 in hematological malignancies are well-discussed.
In general, the effects of both PAX5 and PAX6 on breast cancer have been well addressed. I suggest you add your comments on what could be the reasons for PAX5 playing both proliferative and anti-proliferative roles in breast cancer. I suggest that these different results be indicated as bias or incorrectly designed studies or whether it may play different roles in different histological breast cancers on different mechanisms. With bioinformatic analysis, it can be evaluated in favor of playing a proliferative role as it is higher in patients with breast cancer compared to those without.
While it was found that ‘’PAX8-PPARγ expression was found to correlate with key clinical parameters’’, in your bioinformatic analysis, ‘’a significant decrease in PAX8 expression was observed in both primary tumors and metastatic lesions compared to normal thyroid tissues’’, how do you interpret this contradictory result?
The role of PAX2 and PAX8 in urogenital tract cancers is clearly explained
Rephrase ‘In mediastinal biopsies, PAX8 could be useful to discriminate the origin of cancer cells’ as mediastinal mass biopsies or similar.
I suggest you edit ‘Other PAXS’ to ‘Other PAX’s’
Best regards
Author Response
Reviewer#2
The role of genomic changes and expressed proteins in carcinogenesis and their prognostic importance are the subjects of many studies. I congratulate the authors for their article that generally addresses the PAX family in all cancers.
The article is generally well-written, written in a language that provides a pleasure to read and contains scientific richness. I believe that it will contribute to the literature. My general views and suggestions regarding the study are as follows.
Figures and tables are used appropriately and explanatorily throughout the study.
In the first section, the functions of PAX genes and proteins, their subgroups are explained appropriately and supported by a table.
In the second section, the role and mechanism of PAX-5 in hematological malignancies are well-discussed.
The role of PAX2 and PAX8 in urogenital tract cancers is clearly explained
Reply: We sincerely thank the reviewer for the encouraging and thoughtful feedback. We truly appreciate your positive evaluation of our manuscript, particularly your recognition of the scientific relevance, clarity of writing, and the usefulness of the figures and tables. We are especially grateful for your remarks regarding the overall structure and contribution of the review to the literature on the role of PAX proteins in carcinogenesis. Your supportive comments are greatly motivating and have further reinforced our commitment to delivering a scientifically sound and well-organized article. Thank you again for your kind words and constructive insights.
When mentioning the subgroups of PAX genes and proteins, the types of subgroups are stated in parentheses more than once, I think it would be sufficient to state it once.
Was ‘II.1’ used incorrectly in the expression II.1 PAX5 (paired box 5), used on line 87? Check please.
Reply: we corrected this
In general, the effects of both PAX5 and PAX6 on breast cancer have been well addressed. I suggest you add your comments on what could be the reasons for PAX5 playing both proliferative and anti-proliferative roles in breast cancer. I suggest that these different results be indicated as bias or incorrectly designed studies or whether it may play different roles in different histological breast cancers on different mechanisms. With bioinformatic analysis, it can be evaluated in favor of playing a proliferative role as it is higher in patients with breast cancer compared to those without.
Reply: We thank the reviewer for this insightful and constructive suggestion. We agree that the dual role of PAX5 in breast cancer—as both a tumor suppressor and a potential oncogenic factor—warrants further discussion. In response, we have added a paragraph in the relevant section to address potential reasons behind these seemingly contradictory findings. Specifically, we highlight the possibility that PAX5 may exert different effects depending on breast cancer subtype, tumor microenvironment, and molecular context, rather than attributing discrepancies solely to methodological bias. We also expanded our discussion on how bioinformatic data showing higher PAX5 expression in breast tumors compared to normal tissues could support a context-dependent pro-proliferative role.
While it was found that ‘’PAX8-PPARγ expression was found to correlate with key clinical parameters’’, in your bioinformatic analysis, ‘’a significant decrease in PAX8 expression was observed in both primary tumors and metastatic lesions compared to normal thyroid tissues’’, how do you interpret this contradictory result?
Reply: This discrepancy can be attributed to several factors. First, bulk RNA-sequencing data, such as those in TCGA, do not discriminate between wild-type PAX8 transcripts and those derived from PAX8 fusion events. Second, PAX8-PPARγ fusions are most commonly found in follicular thyroid carcinoma, a more differentiated subtype, whereas the TCGA cohort includes a broader spectrum of thyroid cancers, including less differentiated or anaplastic types in which PAX8 expression is often lost. Lastly, post-transcriptional regulation and fusion protein stability may result in increased protein levels despite reduced gene expression. Thus, while the fusion protein may drive oncogenic behavior in a subset of tumors, the overall PAX8 transcript level in diverse thyroid cancer samples may not reflect this specific mechanism.
Rephrase ‘In mediastinal biopsies, PAX8 could be useful to discriminate the origin of cancer cells’ as mediastinal mass biopsies or similar.
Reply: done
I suggest you edit ‘Other PAXS’ to ‘Other PAX’s’
Reply: done
Round 2
Reviewer 1 Report
Comments and Suggestions for Authors
The authors have addressed my previous comments.